# Computational Retinal Microvascular Biomarkers from an OCTA Image in Clinical Investigation

**DOI:** 10.3390/biomedicines12040868

**Published:** 2024-04-15

**Authors:** Bingwen Lu, Yiming Li, Like Xie, Kin Chiu, Xiaofeng Hao, Jing Xu, Jie Luo, Pak-Chung Sham

**Affiliations:** 1Department of Ophthalmology, School of Clinical Medicine, Li Ka Shing Faculty of Medicine, The University of Hong Kong, Hong Kong SAR, China; 1988willa@sina.com; 2Department of Ophthalmology, Eye Hospital, China Academy of Chinese Medical Sciences, Beijing 100040, China; 3Department of Psychiatry, School of Clinical Medicine, Li Ka Shing Faculty of Medicine, The University of Hong Kong, Hong Kong SAR, China; kestrel614@gmail.com (Y.L.); pcsham@hku.hk (P.-C.S.); 4Centre for PanorOmic Sciences, The University of Hong Kong, Hong Kong SAR, China; 5State Key Laboratory of Brain and Cognitive Sciences, The University of Hong Kong, Hong Kong SAR, China; 6Department of Psychology, The University of Hong Kong, Hong Kong SAR, China

**Keywords:** computational retinal microvasculature biomarkers (CRMB), optical coherence tomography angiography (OCTA), retinal imaging, retinal vein occlusion

## Abstract

Retinal structural and functional changes in humans can be manifestations of different physiological or pathological conditions. Retinal imaging is the only way to directly inspect blood vessels and their pathological changes throughout the whole body non-invasively. Various quantitative analysis metrics have been used to measure the abnormalities of retinal microvasculature in the context of different retinal, cerebral and systemic disorders. Recently developed optical coherence tomography angiography (OCTA) is a non-invasive imaging tool that allows high-resolution three-dimensional mapping of the retinal microvasculature. The identification of retinal biomarkers from OCTA images could facilitate clinical investigation in various scenarios. We provide a framework for extracting computational retinal microvasculature biomarkers (CRMBs) from OCTA images through a knowledge-driven computerized automatic analytical system. Our method allows for improved identification of the foveal avascular zone (FAZ) and introduces a novel definition of vessel dispersion in the macular region. Furthermore, retinal large vessels and capillaries of the superficial and deep plexus can be differentiated, correlating with retinal pathology. The diagnostic value of OCTA CRMBs was demonstrated by a cross-sectional study with 30 healthy subjects and 43 retinal vein occlusion (RVO) patients, which identified strong correlations between OCTA CRMBs and retinal function in RVO patients. These OCTA CRMBs generated through this “all-in-one” pipeline may provide clinicians with insights about disease severity, treatment response and prognosis, aiding in the management and early detection of various disorders.

## 1. Introduction

Retinal structural and functional changes in humans can be manifestations of different physiological or pathological conditions, including retinal disorders such as diabetic retinopathy (DR), retinal vein occlusion (RVO), and age-related macular degeneration (AMD), as well as systemic diseases such as hypertension and arteriolosclerosis [1,2]. Also, as retinal and cerebral vasculatures share many similarities, the retinal vasculature could serve as a mirror reflecting the abnormalities of the cerebral vasculature in neurological diseases [3]. With improved longevity and stress-filled lifestyles, the number of patients suffering from these vision-threatening conditions is steadily increasing. Therefore, there exists an urgent need for a large-scale improvement in the way in which these diseases can be screened, diagnosed and treated.

Retinal imaging is the only way to directly inspect blood vessels and their pathological changes throughout the whole body non-invasively. Various quantitative metrics have been widely used by either automated or semi-automated analysis methods to measure the abnormalities of retinal microvasculature in the context of different retinal, cerebral and systemic disorders [4,5,6]. The development of new retinal biomarkers for the diagnosis and follow-up of patients with retinal, cerebral, as well as systemic disorders is crucial.

Optical coherence tomography angiography (OCTA) is a non-invasive retinal imaging tool that can detect retinal microvasculature in detail [7]. The rapidly developing OCTA technique bridges the gaps between conventional imaging tools in the eye and the brain, adding additional information about the macro- and micro-vasculatures under various circumstances [8,9]. Moreover, due to its high resolution, depth-resolved images of superficial, deep vessel plexus and choriocapillaris can be detected, thus retinal microvasculature can be displayed on separate layers, expanding our knowledge on the organization of the retinal vessels [10].

Current commercially available OCTA devices are not able to provide a wide field of quantifiable parameters of the retinal vasculature due to the limitations of their analyzing software. Therefore, we provided a method for establishing computational retinal microvascular biomarkers (CRMBs) through a knowledge-driven computerized automatic analytical system based on fractal analysis using OCTA images. Our method allows for improved identification of the foveal avascular zone (FAZ) and a novel definition of vessel dispersion in the macular region. Moreover, retinal large vessels and retinal capillaries of the superficial and deep plexus were differentiated concerning retinal pathology. The diagnostic power of OCTA CRMBs and their correlations with retinal function were studied through a cross-sectional study involving 30 healthy subjects and 43 retinal vein occlusion (RVO) patients.

We envisage that these indicative retinal biomarkers generated through this “all-in-one” pipeline would directly lead to effective triage of patients, facilitating early diagnosis, timely treatment, and improved quality of life.

## 2. Materials and Methods

### 2.1. Patients

Patients were recruited from January 2016 to December 2019 at Eye Hospital, China Academy of Chinese Medical Sciences, Beijing, China. All the patients were diagnosed based on color fundus examination and fluorescein angiography (FA) findings by two retinal specialists (Dr. Xie and Dr. Lu). Patients’ demographics and baseline characteristics were described in Table 1. At their initial visit, a confirmed history of RVO with retinal hemorrhage and macular edema (ME) extending to the fovea was present in each enrolled patient. The ischemic type of RVO was defined when the non-perfused area was larger than 5-disc areas on FA in branch retinal vein occlusion (BRVO) patients and when the non-perfused area was larger than 10-disc areas on FA in central retinal vein occlusion (CRVO) patients. All participants received a single intravitreal conbercept (Lumitin; Kang Hong Biotech Co., Ltd., Chengdu, Sichuan, China) (0.5 mg/0.05 mL) injection for ME secondary to RVO. At the initial visit and after the resolution of RVO ME, each patient underwent a comprehensive ophthalmologic examination, including BCVA measurement, slit-lamp biomicroscopy, color fundus photography, and spectral domain OCTA examination on the same day. BCVA was measured using a standard logarithmic visual acuity chart; 45° digital fundus photography (TRC-50LX, Topcon, Tokyo, Japan) was performed after pupil dilation; and OCTA examination was performed using RTVue XR Avanti (RTVue XR Avanti, AngioVue; Optovue, Inc., Fremont, CA, USA). FA was performed if necessary. The inclusion criteria were as follows: no current ME after anti-VEGF treatment (foveal thickness [FT] < 300 µm and no retinal cysts), and OCTA images with good quality (signal strength index > 50). Exclusion criteria included patients with retinal arterial occlusion, previous retinal surgery, history of uveitis, ocular trauma, or any previous or combined retinal diseases that could affect the explanation of OCTA measurements, such as severe diabetic retinopathy, age-related macular degeneration, glaucoma, retinitis pigmentosa and pathologic myopia (>8 dioptres). Moreover, subjects with OCTA images of poor quality for evaluation due to poor fixation, a low signal strength index (<50), or significant media opacity were excluded from the study.

### 2.2. Statistics

Pairwise Kendall’s rank correlation tests were carried out to assess the association among different CRMBs. Correlations with a Bonferonni-corrected *p*-value of less than 0.05 were considered significant. The distribution of CRMBs in normal individuals and RVO patients was visualized through violin plots. Unpaired *t*-tests were conducted to test for the difference across the two groups. We controlled for multiple testing through the Benjamini-Hochberg (BH) procedure, and significance was declared when the adjusted *p*-value was less than 0.1. Kendall’s rank correlations between initial BCVA and different CRMBs were calculated, and multiple testing was controlled for through the BH procedure. Adjusted *p*-values less than 0.1 were considered significant. We fitted LASSO regression models regressing the presence of RVO and initial BCVA on different CRMBs and covariates, respectively. The continuous variables were normalized before model-fitting, and the penalization parameter for LASSO was selected through 10-fold cross-validation with deviance as the criteria. Predictors with non-zero estimated LASSO coefficients were significant in the prediction.

### 2.3. Study Approval

Ethics approval was granted from the Ethics Committee at the China Academy of Chinese Medical Sciences, and the study was conducted in accordance with the tenets of the Declaration of Helsinki. Written informed consent was obtained from each subject before any study procedures or examinations were performed. This consecutive, cross-sectional, observational study enrolled 43 eyes of 43 patients with resolved RVO-ME after one intravitreal injection of an anti-VEGF agent and 30 healthy subjects who were examined at the Department of Ophthalmology of the Eye Hospital of the China Academy of Chinese Medicine Sciences (Beijing, China) between January 2016 and December 2019.

More information is available in the Appendix A, including the quantification pipeline of our OCTA CRMBs and the cross-sectional study design.

## 3. Results and Discussion

### 3.1. OCTA CRMBs Generation

We developed an automatic pipeline in which CRMBs could be extracted and quantified from machine-generated OCTA reports to characterize retinal microvascular states (Figure 1). Raw OCTA images (Figure 1A) of both superficial capillary plexus (SCP) and deep capillary plexus (DCP) were exported from OCTA devices, from which we extracted the digital vasculature map (DVM) corresponding to the 3 × 3 scan area (Figure 1B). The early Treatment of Diabetic Retinopathy (ETDRS) grid was applied to the DVM (Figure 1C) to define the fovea and parafoveal regions. Parafoveal retinal vessels in the SCP were next distinguished from superficial capillaries (Figure 1). OCTA blood flow metrics, including perfusion density of large vessels (PDL), perfusion density of capillaries in the SCP (PDCS), and perfusion density of capillaries in the DCP (PDCD), were quantified by calculating the average pixel signal intensities. Retinal vascular geometric characteristics (RVGCs) of the SCP, including macular vessel tortuosity (MVT), macular vessel diameter (MVDiam), and macular vessel dispersion (MVDisp), were quantified. Another RVGC, the fractal dimension (FD) of both layers (DCP–FDD, SCP–FDS) was calculated based on binarized OCTA images (Figure 1D) through the box-counting procedure to quantify the structural complexity of retinal microvasculature [11]. To identify the FAZ of both retinal layers more accurately, we approximated it with a polygon formed by connecting positive (blood vessel) pixels nearest to the center of the fovea (Figure 1G). FAZ areas (DCP–FAD, SCP–FAS) were computerized, and FAZ acircularities (DCP–FACD, SCP–FACS) were calculated to assess the irregularity of the shape of the FAZ.

### 3.2. OCTA CRMBs Characterize Retinal Pathology

Through OCTA, we may observe the retinal microvasculature of the superficial, deep, avascular outer retina as well as the choroidal vasculature. Previous measurements of the superficial layer included both retinal capillaries and large vessels (arterioles and venules) [12]. However, the hemodynamic and structural changes under hypoxia in larger vessels and capillaries may differ from each other due to different cell constitutions. For example, retinal arterioles contain smooth muscle cells [13], while retinal capillary walls are composed of endothelial cells and pericytes [14]. Retinal large vessels and capillaries in the superficial layer may be distinguished heuristically according to signal intensities and vessel diameters using our framework. OCTA blood flow metrics of the superficial retinal layer can be calculated separately, improving the discriminative power corresponding to different retinal cellular responses to retinal ischemia. It has been shown that RVGCs, such as diameter, tortuosity, length-to-diameter ratios, and branching angles, play a pivotal role in the early detection of retinal and systemic disorders and are associated with disease severity [15]. Morphological and structural changes of retinal large vessels have previously been shown to offer prognostic values for the prediction of DR [16]. Therefore, we subdivided OCTA CRMBs into four categories, targeting retinal large vessels, superficial capillaries, deep capillaries, and FAZ of the macular region, respectively (Figure 2A). Correlation and hierarchical clustering analyses were performed to further assess the underlying association structure of CRMBs in different categories (Figure 2B). Three clusters with more than one CRMB can be identified: (1) FACS and FACD; (2) MVT, FAS and FAD; as well as (3) PDL, FDS, PDCS, FDD and PDCD. In the third cluster, CRMBs related to superficial/deep capillaries were strongly positively correlated with each other. PDL had a weaker intra-cluster correlation compared to the others and was more positively correlated with deep capillaries. The first cluster, including FACS and FACD, was found to be negatively correlated with deep capillary CRMBs, whereas FAD in the second cluster had significantly negative correlations with capillary CRMBs, especially FDs. In our study, MVDiam and MVT were weakly correlated with any other CRMBs.

### 3.3. OCTA CRMBs Correlate with Retinal Function

RVO is the second most common vision-threatening retinal vascular disease after DR, which may be categorized into central retinal vein occlusion (CRVO) and branch retinal vein occlusion (BRVO) [17]. Different from DR, which is a complex metabolic disorder with numerous complications, RVO is a disorder caused mainly by retinal vein obstructions, leading to retinal ischemia (hypoxia). Macular edema (ME) secondary to RVO is the most common complication and is the main cause of vision loss in RVO [18]. Intravitreal injection of anti-vascular endothelial growth factor (VEGF) agents is currently the first-line therapy to treat RVO-ME [19]. However, patients may suffer a relapse despite a complete resolution of ME after treatment [20]. In such circumstances, retinal microvascular abnormalities as well as macular perfusion status are believed to be involved in the impaired retinal function of those patients, determining the treatment response and prognosis [21]. To illustrate the usefulness of our automated analytical framework, CRMB analyses were carried out in 43 OCTA images from RVO patients with resolved ME after anti-VEGF treatment and 30 OCTA images of healthy subjects. All OCTA images used in the cross-sectional study were taken using RTVue XR Avanti (Optovue, Inc., Fremont, CA, USA), covering a 3 × 3 area centered at the fovea.

### 3.4. OCTA CRMBs as Risk Factors of RVO

As demonstrated in the representative OCTA images in Figure 3A, compared to healthy subjects, reduced PDCS, FDS, PDL and MVDiam, as well as increased FAS, FACS, MVDisp and MVT, were detected in typical RVO patients. Figure 3B shows that in our data set, the CRMBs were distributed differently in normal versus RVO samples. Moreover, there was a significant difference in the means of all the CRMBs except MVDiam and MVT between normal subjects and RVO patients. To identify potential risk factors for RVO, we also conducted LASSO analysis, regressing whether a subject had RVO on CRMBs and various covariates (Figure 3C). Results showed that decreased MVDiam, PDCD and FDD, as well as increased FAS and FACS as evident in OCTA images, may indicate a greater probability of having RVO. Vascular densities all over the scanned area in both superficial and deep retinal layers have been reported to be lower in RVO eyes in previous studies [22]. Our result showed that the vascular density of the deep layer might be a more important predictor of RVO. DCP is a relative watershed zone between the retinal vascular and choroidal circulations and therefore more sensitive to ischemic changes [23]. In addition to vascular density measurements, other geometric parameters that describe microvascular morphology, including MVDiam and FD, were also found to be useful in assessing retinal vascular changes. MVDiam is influenced by physiological and pathological determinants such as blood pressure, blood glucose, body mass index, smoking and atherosclerosis; therefore, it may affect RVO or vice versa, as demonstrated by previous findings [24]. Current parameters generated by commercially available OCTA devices do not include vascular caliber, which is an objective and reproducible biomarker. Previous reports found that the change of FAZ area varied in the SCP of RVO eyes, while the FAZ area was enlarged in the DCP of RVO eyes [22]. In this study, we detected the margin of FAZ by approximating it with a polygon formed by connecting positive (blood vessel) pixels nearest to the center of the fovea. LASSO analysis indicated FAZ-related parameters in the SCP to be predictors of RVO, not those in the DCP. To further confirm our findings, further study with increased sample size and expended RVO subtypes are needed.

### 3.5. OCTA CRMBs Significantly Correlate with Visual Acuity

Correlation analyses were performed to study the association between best-corrected visual acuities (BCVAs) and OCTA CRMBs in all 73 samples (30 healthy, 43 RVO). Initial BCVA was significantly positively correlated with PDL, PDCS, PDCD, FDS and FDD (adjusted *p*-value < 0.1) (Figure 4). This may indicate that reduced blood supply to both retinal large vessels and retinal capillaries contributed to retinal functional impairment. We also found significantly negative correlations between initial BCVA and all the FAZ-related CRMBs (adjusted *p*-value < 0.1), proving the enlargement of FAZ should play a role in the progression of RVO (Figure 4). RVGC abnormalities, more specifically increased MVDisp, were also associated with visual acuity loss. MVDisp reflects the degree of disorganization of the retinal microvasculature, which can be influenced by many possible factors, including loss of vascular microstructural integrity and blood flow impairment [25]. In our study, we defined MVDisp as the degree of centripetalism of the retinal vessels (Appendix A). The larger MVDisp, the fewer centripetal parafoveal vessels there are on average. This novel biomarker might be a useful CRMB in RVO management due to its strong correlation with retinal function and clinical value in the BRVO/CRVO sub-classification. Our LASSO analysis indicated that PDL, PDCS, PDCD, and FACD were significantly predictive of initial BCVA, which was in line with our correlation analysis results.

OCTA has emerged as a novel, fast, safe, and non-invasive imaging technique for analyzing the retinal and choroidal microvasculature in vivo. The technology has significantly expanded our knowledge of the retinal vasculature and provides a unique view of the retinal or choroidal layer. The exact site and severity of abnormal blood flow detected by OCTA plays a pivotal role in the detection and monitoring of a wide range of disorders. Therefore, OCTA plays a pivotal role in the detection and monitoring of a wide range of disorders. In our study, we designed and developed an automatic, knowledge-driven framework for quantifying different CRMBs from raw OCTA images, introducing a novel definition of MVDisp and an improved method for FAZ demarcation. The strength of our study lies in our capability to distinguish retinal large vessels from retinal capillaries of both the superficial and deep plexus. Partially based on this large vessel/capillary separation, we categorized OCTA CRMBs into four classes in accordance with retinal pathology. By applying our methodology to a cross-sectional study regarding RVO, we have demonstrated that our CRMBs correlate well with retinal functions like visual acuity and may be risk factors for the development of RVO. Incorporation of OCTA CRMBs into retinal imaging analyses may aid scientists and ophthalmologists in the study, management, and early diagnosis of retinal and systemic disorders.

As mentioned before, retinal structural and functional changes in humans can be manifestations of different physiological or pathological conditions. Over the years, various studies have investigated the relationship between vascular diameters and risk factors for cardiovascular diseases, diabetes, and chronic kidney disease from fundus images [26]. Most recently, Al-Nofal et al. have demonstrated that (retinal vasculopathy with cerebral leukoencephalopathy and systemic manifestations) RVCL-S, which is a monogenic small vessel disease caused by hererozygous C-terminal truncating mutations in *TREX1*, causes an increase in the size of the FAZ in symptomatic RVCL-S patients compared to healthy participants using OCTA analysis [27]. Bianchetti et al. found that increased erythrocyte membrane fluidity is associated with a higher cardiovascular risk in subjects with type 2 diabetes [28]. The introduction and development of OCTA, providing quick and non-invasive high-resolution angiograms, enables the assessment of retinal microvasculature and is suggested as a potential tool in the early detection of retinal microvascular changes in many systemic disorders. Therefore, our novel definition and quantification method of CRMBs (retinal microvascular parameters in different capillary plexus layers), which showed a close correlation with retinal pathology and function, might be used as new biomarkers for risk assessment in several diseases, as well as clinical decision-making, diagnosis, prevention, follow-up and treatment evaluation of those diseases.

Recently, other researchers have also developed several automated standardized frameworks, driven by the recent advancements in deep learning, to extract OCTA biomarkers for the early diagnosis of several diseases. Xie, J. et al. defined and extracted geometrical parameters of retinal microvasculature at different retinal layers and in the FAZ from segmented OCTA images obtained using well-validated state-of-the-art deep learning models, whose results demonstrated the applicability of OCTA for the diagnosis of Alzheimer’s Disease and mild cognitive impairment [29]. Different quantitative methods might have advantages and disadvantages of their own. Our novel definition starts with retinal pathology, and the results of the improved quantification method showed a great correlation between retinal pathology and function. Further cross-sectional and longitudinal clinical studies with a large sample size need to be performed to determine whether these findings are a reliable method for clinical application.

The major limitation of this pilot study is that the sample size is relatively small, which only involves 73 subjects. The small sample size might impact the generalizability of our findings. Clinical studies with larger samples and further follow-up periods are necessary to further confirm the feasibility of our method. There are several other limitations to this study. Firstly, using the currently available software of AngioAnaytics (2017.1 software version, phase 7.0 update) on the Optovue OCTA machine, the images of the FAZ area (by automated segmentation from the internal limiting membrane to 10 μm below the outer plexiform layer) can be analyzed. We applied a novel definition and improved method for FAZ analysis, which might be more accurate. However, the feasibility of the proposed method should be further studied by a treatment monitoring and statistical analysis process with larger samples, and a comparative analysis should be conducted between the two methods to further confirm the advantage of our method. Secondly, since OCTA demands high levels of patient cooperation and attentiveness, poor-quality images can lead to inaccurate interpretation due to motion artifacts. Thirdly, OCTA CRMBs proved to be useful in the management of RVO patients; the application of those retinal biomarkers should be explored further in other retinal vascular disorders, systemic diseases, and neurological and neuro-ophthalmological diseases. Finally, a cross-sectional study does not measure changes in retinal microvascular parameters over time or disease progression. Future longitudinal studies in larger cohorts must be conducted to accurately interpret the clinical value of OCTA CRMBs in the management and monitoring of diseases.

## Figures and Tables

**Figure 1 biomedicines-12-00868-f001:**
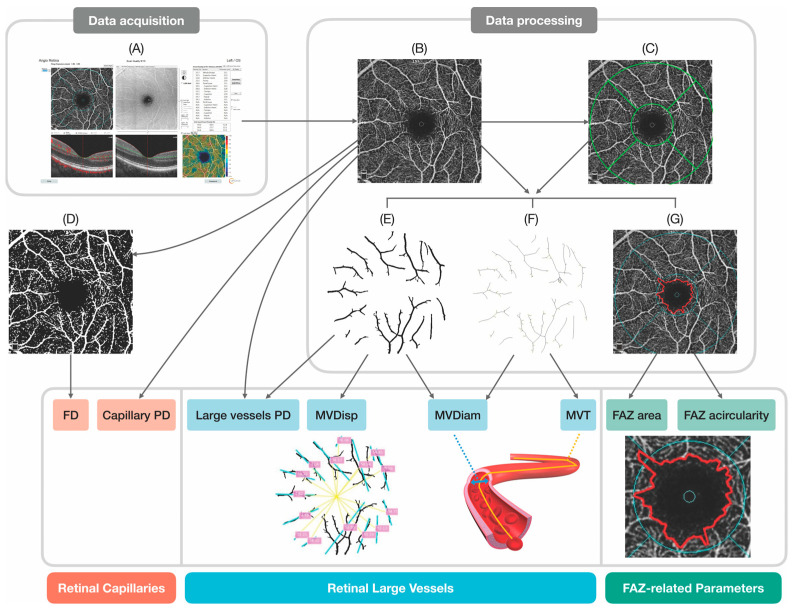
The pipeline of automatic quantification of OCTA CRMBs. (**A**) Raw OCTA image of the SCP; (**B**) Extracted DVM; (**C**) DVM with applied ETDRS grid (green); (**D**) DVM after thresholding; (**E**) Delineated retinal large vessels; (**F**) Skeletonized retinal large vessels; (**G**) Demarcated FAZ (red outline). Abbreviations used—DVM: digital vasculature map; FD: fractal dimension; PD: perfusion density; MVDisp: macular vessel dispersion; MVDiam: macular vessel diameter; MVT: macular vessel tortuosity; FAZ: foveal avascular zone.

**Figure 2 biomedicines-12-00868-f002:**
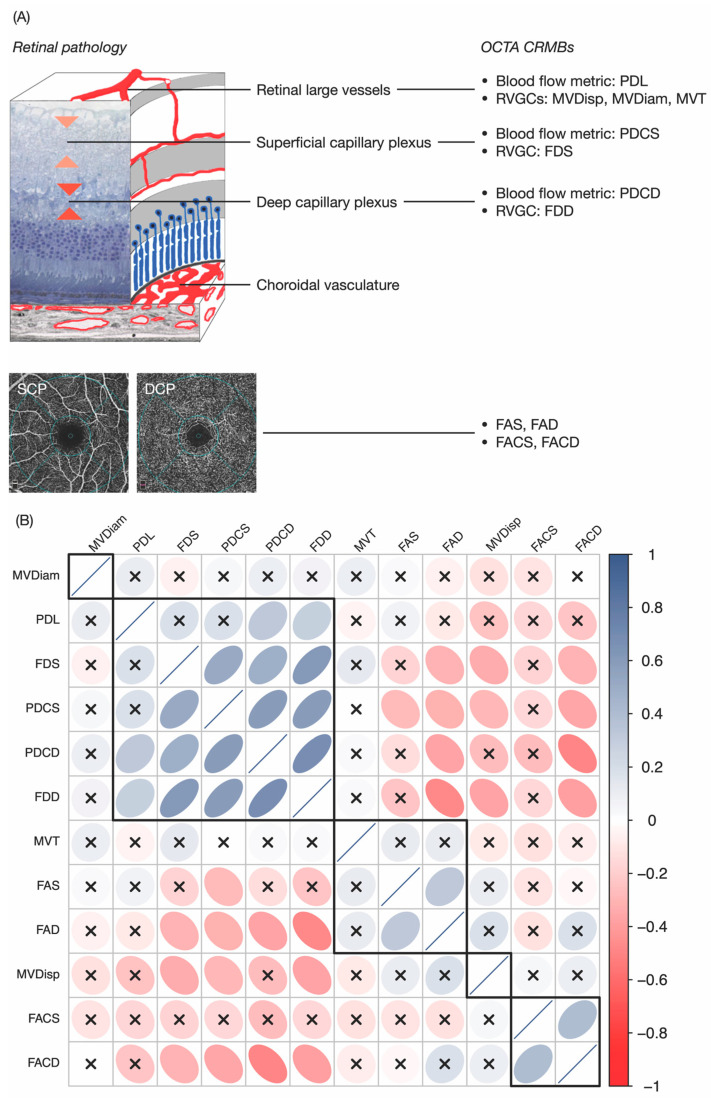
OCTA CRMBs correlate with retinal pathology. (**A**) Blood and nutrients are supplied to the superficial layer mainly by the retinal vasculature (above), while to the deep layer by both the choroidal (below the outer retina) and the retinal vasculature. OCTA CRMBs are classified into four categories accordingly. (**B**) Correlation among different CRMBs (*n* = 73). The heatmap was colored pairwise by Kendall’s rank correlations. Red squares indicate negative correlations, and blue squares represent the inverse. Color intensity is proportional to the magnitude of correlation and crosses indicate insignificant correlations (Bonferroni-corrected *p*-value < 0.05). Rows and columns were ordered through hierarchical clustering with complete linkage, and the resulting dendrogram was cut into five subtrees by height. The subtrees were shown as black squares. Abbreviations used—PDL: perfusion density of large vessels; MVDisp: macular vessel dispersion; MVDiam: macular vessel diameter; MVT: macular vessel tortuosity; PDCS/PDCD: superficial/deep perfusion density; FDS/FDD: superficial/deep fractal dimension; FAS/FAD: superficial/deep foveal avascular zone area; FACS/FACD: superficial/deep foveal avascular zone acircularity.

**Figure 3 biomedicines-12-00868-f003:**
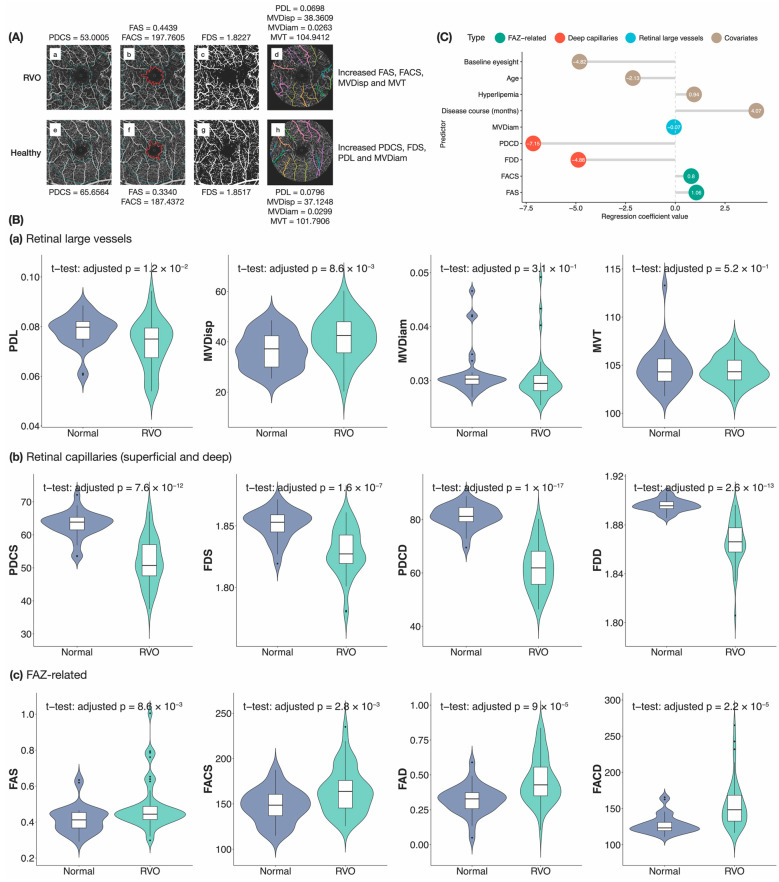
Diagnostic power analysis of CRMBs. (**A**) Typical OCTA images of an RVO patient and a healthy individual. (**a**–**d**) Original DVM, demarcated FAZ, thresholded DVM and delineated retinal large vessels of the RVO patient. (**e**–**h**) Original DVM, demarcated FAZ, thresholded DVM and delineated retinal large vessels of the healthy individual. (**B**) The distribution of CRMBs is different across normal individuals (*n* = 30) and RVO patients (*n* = 43). In the violin plots, horizontal bars represented the median, lower/upper hinges corresponded to the first/third quartiles, and whiskers were plotted through Tukey’s method. Defining the inter-quartile range (IQR) as the distance between the first and third quartiles, the lower/upper whisker extended from the median to the smallest/largest value no further than 1.5 × IQR from the median. Outliers beyond the end of the whiskers were plotted individually. Kernel density plots were included to the sides of the boxplots. *p*-values from unpaired *t*-tests testing if CRMB values differ across the two groups after adjusting for multiple testing were shown. (**C**) Non-zero LASSO coefficients from the model regressing RVO on different CRMBs and covariates (*n* = 73). Regression coefficients of predictors in different categories were sorted in ascending order of their values. Abbreviations used—RVO: retinal vein occlusion; PDL: perfusion density of large vessels; MVDisp: macular vessel dispersion; MVDiam: macular vessel diameter; MVT: macular vessel tortuosity; PDCS/PDCD: superficial/deep perfusion density; FDS/FDD: superficial/deep fractal dimension; FAZ: foveal avascular zone; FAS/FAD: superficial/deep FAZ area; FACS/FACD: superficial/deep FAZ acircularity; BCVA: best-corrected visual acuity.

**Figure 4 biomedicines-12-00868-f004:**
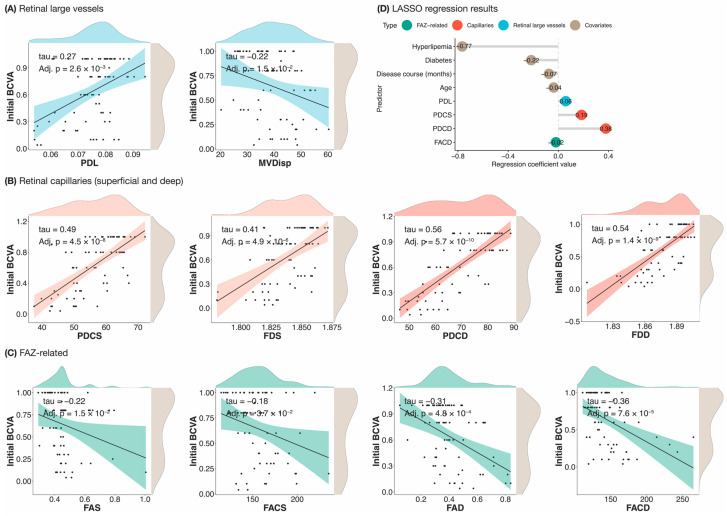
The relationship between initial BCVA and different CRMBs. (**A**–**C**) Significant correlations between initial BCVA and different CRMBs. Only the pairs with a Kendall’s rank correlation significant after controlling for multiple testing (adjusted *p*-value < 0.1) were plotted. A linear regression line with 95% confidence interval (shaded region) was added to each scatterplot. Variable distributions were shown as density plots on the sides of the scatterplots. (**D**) Non-zero LASSO coefficients from the model regressing initial BCVA on different CRMBs and covariates (*n* = 73). Regression coefficients of predictors in different categories were sorted in ascending order of their values. Abbreviations used—tau: Kendall’s rank correlation; Adj. p: adjusted *p*-value; PDL: perfusion density of large vessels; MVDisp: macular vessel dispersion; MVDiam: macular vessel diameter; MVT: macular vessel tortuosity; PDCS/PDCD: superficial/deep perfusion density; FDS/FDD: superficial/deep fractal dimension; FAZ: foveal avascular zone; FAS/FAD: superficial/deep foveal avascular zone area; FACS/FACD: superficial/deep foveal avascular zone acircularity; BCVA: best-corrected visual acuity.

**Table 1 biomedicines-12-00868-t001:** Patient demographics and baseline characteristics.

	RVO	Control	*p*-Value	Adjusted *p*-Value
Number of Patients and Eyes	43 (43 eyes)	30 (30 eyes)	-	-
Sex (men/women)	19/24	17/13	0.2940	0.3360
Study eye (right/left)	22/21	15/15	0.9221	0.9221
Age, years	55.42 ± 12.95	44.74 ± 15.37	0.0029	0.0077
Subtype (BRVO/CRVO)	25/18	-	-	-
Subtype (ischemic/non-ischemic)	11/32	-	-	-
Diabetes mellitus, yes/no	2/41	0/30	0.2310	0.3080
Hypertension, yes/no	15/28	0/30	0.0003	0.0012
Hyperlipidaemia, yes/no	6/37	0/30	0.0327	0.0523
Atherosclerosis, yes/no	11/32	1/29	0.0117	0.0234
Duration from the initial visit, months	2.00 ± 2.06	-	-	-
BCVA at initial visit	0.43 ± 0.29	0.93±0.10	<0.001	<0.001
BCVA at inclusion	0.73 ± 0.28	-	-	-

Continuous variables were expressed as mean ± SD with *p*-values from independent *t*-tests (2 tails), whereas binary variables were expressed as counts with *p*-values from proportion tests. Significant at adjusted *p*-value < 0.05. Abbreviations: RVO, retinal vein occlusion; BRVO, branch retinal vein occlusion; CRVO, central retinal vein occlusion; BCVA, best corrected visual acuity.

## Data Availability

Data are contained within the article and Appendix A.

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
