# Peer review of "Computational Retinal Microvascular Biomarkers from an OCTA Image in Clinical Investigation"

_biomedicines, 2024, doi:10.3390/biomedicines12040868_

Round 1
Reviewer 1 Report
Comments and Suggestions for Authors
The authors provided a framework for extracting computational retinal microvasculature biomarkers (CRMBs) from OCTA images. The detailed comments are as below:
1. The type of the OCTA machine needs to be clarified. SD or SS-OCT? Manufacture and series? It is important since the readers need to know if these quantification could be applied on other OCTA machines.
2. Which scan pattern was used to develop and test this framework? We know the visualization of the vessels is affected by different scan patterns - the larger field of view of a scan, the less detail will be seen.
3. What if the computed results (e.g., FAZ) were incorrect? Does the algorithm include a quality check and an editing function?
4. With the current OCTA machine that the authors used in this study, is any of the retinal quantification already available on the machine?
5. I do not see the supplemental materials. Was it missing?
Author Response
Reviewer 1
The authors provided a framework for extracting computational retinal microvasculature biomarkers (CRMBs) from OCTA images. The detailed comments are as below:
1.The type of the OCTA machine needs to be clarified. SD or SS-OCT? Manufacture and series? It is important since the readers need to know if these quantification could be applied on other OCTA machines.
Response
We thank the reviewer for the helpful suggestion. We have highlighted the OCTA machine we used in the paper as follows: “To illustrate the usefulness of our automated analytical framework, CRMBs analyses were carried out for 43 OCTA images from RVO patients with resolved ME after anti-VEGF treatment and 30 OCTA images of healthy subjects. All OCTA images used in the cross-sectional study were scanned with RTVue XR Avanti (Optovue, Inc., Fremont, CA, USA) covering a 3 × 3 area centered at the fovea.”
2.Which scan pattern was used to develop and test this framework? We know the visualization of the vessels is affected by different scan patterns - the larger field of view of a scan, the less detail will be seen.
Response
We thank the reviewer for the helpful suggestion. We have highlighted the scan pattern we used to develop and test this framework in the paper as follows: “To illustrate the usefulness of our automated analytical framework, CRMBs analyses were carried out for 43 OCTA images from RVO patients with resolved ME after anti-VEGF treatment and 30 OCTA images of healthy subjects. All OCTA images used in the cross-sectional study were scanned with RTVue XR Avanti (Optovue, Inc., Fremont, CA, USA) covering a 3 × 3 area centered at the fovea.” In order to include more details of retinal microvasculature, we chose the smaller field of view (3 × 3 area ) in this study.
3.What if the computed results (e.g., FAZ) were incorrect? Does the algorithm include a quality check and an editing function?
Response
We thank the reviewer for the helpful suggestion. Using the software of AngioAnaytics which is now available on the Optovue OCTA machine, the images of FAZ area (by automated segmentation from internal limiting membrane to 10μm below the outer plexiform layer) can be analyzed. However, we used a novel definition and improved method for FAZ analysis, which might be more accurate. The method was describe in the paper as follows: “To identify the FAZ of both retinal layers more accurately, we approximated it with a polygon formed by connecting positive (blood vessel) pixels nearest to the center of the fovea (Fig 1G). FAZ areas (DCP – FAD, SCP – FAS) were computerized and FAZ acircularities (DCP – FACD, SCP – FACS) were calculated to assess the irregularity of the shape of the FAZ.”. The feasibility of the proposed method should be further studied by a treatment monitoring and statistical analysis process with larger samples. Also, in view of your helpful suggestion, we considered it meaningful to conduct a comparative analysis between the two methods. This was one of our limitations in this study and we included this part in our further study. We revised the limitation part accordingly: “Firstly, using the currently available software of AngioAnaytics on the Optovue OCTA machine, the images of FAZ area (by automated segmentation from internal limiting membrane to 10μm below the outer plexiform layer) can be analyzed. We applied a novel definition and improved method for FAZ analysis, which might be more accurate. However, The feasibility of the proposed method should be further studied by a treatment monitoring and statistical analysis process with larger samples, and a comparative analysis should be conducted between the two methods to further confirm the advantage of our method. ”.
4.With the current OCTA machine that the authors used in this study, is any of the retinal quantification already available on the machine?
Response
We thank the reviewer for the helpful suggestion. The images of FAZ area (by automated segmentation from internal limiting membrane (ILM) to 10 μm below the outer plexiform layer (OPL0), SCP (by automated segmentation from ILM to external boundary of ganglion cell layer), and DCP (by automated segmentation from inner plexiform layer to OPL) were analyzed with the software of AngioAnalytics. Vascular density was calculated as a percentage of the area occupied by vessels in selecting area. However, the goal of our study was to provide a framework for extracting CRMBs from OCTA images through a knowledge-driven computerized automatic analytical system. Different from the currently available automatic method, our method allows for improved identification of FAZ and introduces a novel definition of vessel dispersion in the macular region. Also, retinal large vessels and capillaries of superficial and deep plexus can be differentiated, correlating with retinal pathology.
- I do not see the supplemental materials. Was it missing?
Response
We thank the reviewer for the helpful suggestion. The supplemental materials were missing in the first submission, and we have successfully loaded the supplemental materials in the re-submission this time. Sorry for the missing.

Reviewer 2 Report
Comments and Suggestions for Authors
This study makes a significant contribution to the field of retinal imaging, particularly in the application of OCTA for identifying retinal microvascular biomarkers. However, there are several areas where the manuscript could be strengthened:
1) Study's sample size is relatively small, involving 73 subjects. This limitation could impact the generalizability of your findings. Future studies with a larger and more diverse population would be beneficial to validate and extend your results.
2) The statistical analysis in your study, particularly the correlations between best-corrected visual acuities (BCVAs) and OCTA CRMBs, requires a more in-depth examination. While your initial findings indicate significant correlations, there are several aspects that warrant further exploration to enhance the statistical rigor of your study:
-It is crucial to assess and control for potential confounding variables that might influence the observed correlations. Factors such as age, duration of the disease, and presence of other systemic conditions can affect both BCVA and retinal microvascular changes. Multivariate analyses that include these potential confounders would provide a clearer understanding of the true relationship between BCVAs and CRMBs.
-
- Verify that the data meets the assumptions of the statistical tests used. For instance, if Pearson's correlation was used, ensure that the data is normally distributed. If not, consider non-parametric alternatives like Spearman's rank correlation.
-
- While statistical significance is important, the effect size and its clinical relevance are equally crucial. It would be beneficial to discuss how the observed correlations translate into clinical practice. Are the changes in biomarkers significant enough to impact clinical decisions or patient outcomes?
-
- Given the relatively small sample size, it's important to address the statistical power of your study. A post-hoc power analysis could help in understanding if the sample size was sufficient to detect the observed effects. Additionally, a sample size justification, based on prior studies or preliminary data, would strengthen the validity of your results.
3)You suggest that OCTA CRMBs could be useful in managing other retinal vascular disorders, systemic diseases, and neurological conditions. However, I raise concerns about the specificity of these biomarkers for different diseases. A more nuanced discussion on the specificity and potential limitations of applying these biomarkers across various conditions would be beneficial.
4)In the introduction and in the discussion I recommend a thorough comparison of your findings with similar studies that have employed OCTA for biomarker identification. This is crucial for situating your research within the broader scientific context. For example:
-
Xie, J., et al. (2023) in the British Journal of Ophthalmology discuss the use of OCTA in Alzheimer's Disease and mild cognitive impairment (https://bjo.bmj.com/content/early/2023/01/03/bjo-2022-321399).
-
A study from Frontiers in Neurology (Front. Neurol., 26 August 2022 Sec. Neurological Biomarkers) explores OCTA biomarkers in RVCL-S patients (https://www.frontiersin.org/articles/10.3389/fneur.2022.989536/full).
5)While these comparisons could provide valuable insights, I also have concerns regarding the specificity of the OCTA-derived biomarkers in your study. Given the diverse pathologies and etiologies in retinal and systemic diseases, it's crucial to address whether these biomarkers are specific to particular conditions or if they might present in a range of disorders. A detailed discussion on this aspect would greatly enhance the manuscript, providing clarity on the limitations and applicative scope of your findings.
6) Additionally, it would be beneficial to discuss the study by Bianchetti et al. (2020), which explores erythrocyte membrane fluidity as a marker of diabetic retinopathy in type 1 diabetes (https://onlinelibrary.wiley.com/doi/10.1111/eci.13455). This work employs a different technique, and discussing potential functional correlations between their findings and your OCTA-derived biomarkers could offer valuable insights into the broader implications of biomarker identification in retinal diseases.
Comments on the Quality of English LanguageN/A
Author Response
Reviewer 2
This study makes a significant contribution to the field of retinal imaging, particularly in the application of OCTA for identifying retinal microvascular biomarkers. However, there are several areas where the manuscript could be strengthened:
1.Study's sample size is relatively small, involving 73 subjects. This limitation could impact the generalizability of your findings. Future studies with a larger and more diverse population would be beneficial to validate and extend your results.
Response
We thank the reviewer for the helpful suggestion. The sample size of our pilot study which only involved 73 subjects was our major limitation, which could impact the results of our findings. The feasibility of the proposed method should be further studied by a treatment monitoring and statistical analysis process with larger samples. We have been working on that in our further consecutive study. We revised the limitation part accordingly: “The major limitation in this pilot study is that the sample size is relatively small, which only involves 73 subjects. The small sample size might impact the generalizability of our findings. Clinical studies with larger samples and further follow-up periods are necessary to further confirm the feasibility of our method. ”.
- The statistical analysis in your study, particularly the correlations between best-corrected visual acuities (BCVAs) and OCTA CRMBs, requires a more in-depth examination. While your initial findings indicate significant correlations, there are several aspects that warrant further exploration to enhance the statistical rigor of your study:
2.1 It is crucial to assess and control for potential confounding variables that might influence the observed correlations. Factors such as age, duration of the disease, and presence of other systemic conditions can affect both BCVA and retinal microvascular changes. Multivariate analyses that include these potential confounders would provide a clearer understanding of the true relationship between BCVAs and CRMBs.
Response
We thank the reviewer for the helpful suggestion. We have now fitted a LASSO model regressing the initial BCVA on different CRMBs and covariates including age, disease course in months, and other systemic conditions (hypertension, diabetes, hyperlipemia, and atherosclerosis). We found that after including the potential confounders, the CRMBs PDL, PDCS, PDCD, and FACD were predictive of initial BCVA. We have updated the manuscript with the results for the new set of analyses and added Fig. 4D.
2.2 Verify that the data meets the assumptions of the statistical tests used. For instance, if Pearson's correlation was used, ensure that the data is normally distributed. If not, consider non-parametric alternatives like Spearman's rank correlation.
Response
We thank the reviewer for the insightful advice. We have now switched to Kendall’s rank correlation for all the correlation analyses. Our previous results and conclusions still stand after the change. The manuscript and the related figures (Fig. 2B and Fig. 4A-C) have been revised accordingly.
2.3 While statistical significance is important, the effect size and its clinical relevance are equally crucial. It would be beneficial to discuss how the observed correlations translate into clinical practice. Are the changes in biomarkers significant enough to impact clinical decisions or patient outcomes?
Response
We thank the reviewer for pointing out the need for clarification. Our major goal of the study is to find a method that could generate OCTA biomarkers that correlates well between retinal pathology and function. We tested the application possibility of our method in RVO patients just because those geometric biomarkers could help clinical doctors in their decision-making process such as whether to consider anti-VEGF injection, or whether the macular edema would recur and when. Therefore, our further studies have been working on the longitudinal follow-up investigations to observe the changes of those CRMBs during disease progression.
2.4 Given the relatively small sample size, it's important to address the statistical power of your study. A post-hoc power analysis could help in understanding if the sample size was sufficient to detect the observed effects. Additionally, a sample size justification, based on prior studies or preliminary data, would strengthen the validity of your results.
Response
We thank the reviewer for the advice. We did not perform post-hoc power analysis due to several papers we refereed to (https://www.tandfonline.com/doi/abs/10.1198/000313001300339897). Since this was our pilot perspective cross-sectional study, we did not include the sample size justification. We just collected all the available samples we could get at that time for our study which was our limitation.
3.You suggest that OCTA CRMBs could be useful in managing other retinal vascular disorders, systemic diseases, and neurological conditions. However, I raise concerns about the specificity of these biomarkers for different diseases. A more nuanced discussion on the specificity and potential limitations of applying these biomarkers across various conditions would be beneficial.
3.1 In the introduction and in the discussion I recommend a thorough comparison of your findings with similar studies that have employed OCTA for biomarker identification. This is crucial for situating your research within the broader scientific context. For example:
Xie, J., et al. (2023) in the British Journal of Ophthalmology discuss the use of OCTA in Alzheimer's Disease and mild cognitive impairment (https://bjo.bmj.com/content/early/2023/01/03/bjo-2022-321399).
A study from Frontiers in Neurology (Front. Neurol., 26 August 2022 Sec. Neurological Biomarkers) explores OCTA biomarkers in RVCL-S patients (https://www.frontiersin.org/articles/10.3389/fneur.2022.989536/full).
Response
We thank the reviewer for the insightful advice. We have discussed the paper you suggested in the Discussion part as follows: “As mentioned before, retinal structural and functional changes in humans can be manifestations of different physiological or pathological conditions. Over the years, various studies investigated the relationship between vascular diameters and risk factors for cardiovascular diseases, diabtetes, and chronic kidney disease from fundus images [26]. Most recently, Al-Nofal et al have demonstrated that (retinal vasculopathy with cerebral leukoencephalopathy and systemic manifestations) RVCL-S, which is a monogenic small vessel disease caused by hererozygous C-terminal truncating mutations in TREX1, causes an increase in the size of the FAZ in symptomatic RVCL-S patients compared to healthy participants using OCTA analysis [27]. Bianchetti et al found that increased erythrocyte membrane fluidity is associated with a higher cardiovascular risk in subjects with type 2 diabetes [28]. The introduction and development of OCTA, providing quick and non-invasive high-resolution angiograms, enables the assessment of retinal microvasculature and is suggested as a potential tool in the early detection of retinal microvascular changes in many systemic disorders. Therefore, our novel definition and quantification method of CRMBs (retinal microvascular parameters in different capillary plexus layers) which showed close correlation with retinal pathology and function might be used as new biomarkers for risk assessment in several diseases, as well as clinical decision-making, diagnosis, prevention, follow-up and treatment evaluation of those diseases.
Recently, other researchers have also developed several automated standardized framework, driven by the recent advancements in deep learning to extract OCTA biomarkers for the earl diagnosis of several diseases. Xie, J., et al defined and extracted geometrical parameters of retinal microvasculature at different retinal layers and in the FAZ from segmented OCTA images obtained using well-validated state-of-the-art deep learning models, whose results demonstrated the applicability of OCTA for the diagnosis of Alzheimer’s Disease and mild cognitive impairment [29]. Different quantitative methods might have advantages and disadvantages of its own. Our novel definition starts form retinal pathology and the results of the improved quantification method showed great correlation between retinal pathology and function. Further cross-sectional and longitudinal clinical studies with large sample size need to be performed to determine whether these finding are a reliable method for clinical application.”.
3.2 While these comparisons could provide valuable insights, I also have concerns regarding the specificity of the OCTA-derived biomarkers in your study. Given the diverse pathologies and etiologies in retinal and systemic diseases, it's crucial to address whether these biomarkers are specific to particular conditions or if they might present in a range of disorders. A detailed discussion on this aspect would greatly enhance the manuscript, providing clarity on the limitations and applicative scope of your findings.
Response
We thank the reviewer for the insightful advice. We have discussed the paper you suggested in the Discussion part as follows: “As mentioned before, retinal structural and functional changes in humans can be manifestations of different physiological or pathological conditions. Over the years, various studies investigated the relationship between vascular diameters and risk factors for cardiovascular diseases, diabtetes, and chronic kidney disease from fundus images [26]. Most recently, Al-Nofal et al have demonstrated that (retinal vasculopathy with cerebral leukoencephalopathy and systemic manifestations) RVCL-S, which is a monogenic small vessel disease caused by hererozygous C-terminal truncating mutations in TREX1, causes an increase in the size of the FAZ in symptomatic RVCL-S patients compared to healthy participants using OCTA analysis [27]. Bianchetti et al found that increased erythrocyte membrane fluidity is associated with a higher cardiovascular risk in subjects with type 2 diabetes [28]. The introduction and development of OCTA, providing quick and non-invasive high-resolution angiograms, enables the assessment of retinal microvasculature and is suggested as a potential tool in the early detection of retinal microvascular changes in many systemic disorders. Therefore, our novel definition and quantification method of CRMBs (retinal microvascular parameters in different capillary plexus layers) which showed close correlation with retinal pathology and function might be used as new biomarkers for risk assessment in several diseases, as well as clinical decision-making, diagnosis, prevention, follow-up and treatment evaluation of those diseases.
Recently, other researchers have also developed several automated standardized framework, driven by the recent advancements in deep learning to extract OCTA biomarkers for the earl diagnosis of several diseases. Xie, J., et al defined and extracted geometrical parameters of retinal microvasculature at different retinal layers and in the FAZ from segmented OCTA images obtained using well-validated state-of-the-art deep learning models, whose results demonstrated the applicability of OCTA for the diagnosis of Alzheimer’s Disease and mild cognitive impairment [29]. Different quantitative methods might have advantages and disadvantages of its own. Our novel definition starts form retinal pathology and the results of the improved quantification method showed great correlation between retinal pathology and function. Further cross-sectional and longitudinal clinical studies with large sample size need to be performed to determine whether these finding are a reliable method for clinical application.”.
3.3 Additionally, it would be beneficial to discuss the study by Bianchetti et al. (2020), which explores erythrocyte membrane fluidity as a marker of diabetic retinopathy in type 1 diabetes (https://onlinelibrary.wiley.com/doi/10.1111/eci.13455). This work employs a different technique, and discussing potential functional correlations between their findings and your OCTA-derived biomarkers could offer valuable insights into the broader implications of biomarker identification in retinal diseases.
Response
We thank the reviewer for the insightful advice. We have discussed the paper you suggested in the Discussion part as follows: “Bianchetti et al found that increased erythrocyte membrane fluidity is associated with a higher cardiovascular risk in subjects with type 2 diabetes [28]. The introduction and development of OCTA, providing quick and non-invasive high-resolution angiograms, enables the assessment of retinal microvasculature and is suggested as a potential tool in the early detection of retinal microvascular changes in many systemic disorders. Therefore, our novel definition and quantification method of CRMBs (retinal microvascular parameters in different capillary plexus layers) which showed close correlation with retinal pathology and function might be used as new biomarkers for risk assessment in several diseases, as well as clinical decision-making, diagnosis, prevention, follow-up and treatment evaluation of those diseases. ”.

Round 2
Reviewer 1 Report
Comments and Suggestions for Authors
The authors addressed my concerns.
Reviewer 2 Report
Comments and Suggestions for Authors
Can be accepted
Comments on the Quality of English Languageok